# *Teucrium polium*: Potential Drug Source for Type 2 Diabetes Mellitus

**DOI:** 10.3390/biology11010128

**Published:** 2022-01-13

**Authors:** Yaser Albadr, Andrew Crowe, Rima Caccetta

**Affiliations:** Curtin Medical School, Curtin Health Innovation Research Institute, Faculty of Health Sciences, Curtin University, Bentley, WA 6102, Australia; yaser.albadr@postgrad.curtin.edu.au (Y.A.); A.P.Crowe@curtin.edu.au (A.C.)

**Keywords:** glucose lowering, insulin secretion, phenolic compounds, flavonoids, anti-diabetic

## Abstract

**Simple Summary:**

*Teucrium polium* (also known as Golden Germander) is a herb brewed and drunk as a tea by the locals of the Mediterranean region, used mostly to treat a number of illnesses including diabetes. When consumed regularly, the tea can be problematic since some of its ingredients can be toxic or interfere with other medications taken by the patient. Current anti-diabetic medications are not always suitable nor optimal for all patients living with diabetes and therefore new drugs are constantly being sought after which may be more useful and/or present less side effects. Therefore, identifying the specific constituents that give the desired anti-diabetic effect, isolating them and developing them further may provide new useful anti-diabetic drugs. This paper discusses some key compounds found in Golden Germander that might be valuable for developing a new medication for type 2 diabetics whilst outlining some issues with the research conducted thus far.

**Abstract:**

The prevalence of type 2 diabetes mellitus is rising globally and this disease is proposed to be the next pandemic after COVID-19. Although the cause of type 2 diabetes mellitus is unknown, it is believed to involve a complex array of genetic defects that affect metabolic pathways which eventually lead to hyperglycaemia. This hyperglycaemia arises from an inability of the insulin-sensitive cells to sufficiently respond to the secreted insulin, which eventually results in the inadequate secretion of insulin from pancreatic β-cells. Several treatments, utilising a variety of mechanisms, are available for type 2 diabetes mellitus. However, more medications are needed to assist with the optimal management of the different stages of the disease in patients of varying ages with the diverse combinations of other medications co-administered. Throughout modern history, some lead constituents from ancient medicinal plants have been investigated extensively and helped in developing synthetic antidiabetic drugs, such as metformin. *Teucrium polium* L. (*Tp*) is a herb that has a folk reputation for its antidiabetic potential. Previous studies indicate that *Tp* extracts significantly decrease blood glucose levels *r* and induce insulin secretion from pancreatic β-cells in vitro. Nonetheless, the constituent/s responsible for this action have not yet been elucidated. The effects appear to be, at least in part, attributable to the presence of selected flavonoids (apigenin, quercetin, and rutin). This review aims to examine the reported glucose-lowering effect of the herb, with a keen focus on insulin secretion, specifically related to type 2 diabetes mellitus. An analysis of the contribution of the key constituent flavonoids of *Tp* extracts will also be discussed.

## 1. Introduction

Diabetes mellitus (DM) is defined as a chronic and multifunctional metabolic disorder characterised by hyperglycaemia, resulting from impaired insulin secretion, insulin action or both. These pathophysiological processes lead to malfunctions in carbohydrate, protein and fat metabolism. The two main types of DM are (i) type 1, which mainly results from autoimmune destruction of the pancreatic β-cells leading to a lack of insulin secretion and (ii) type 2, the most prevalent, that encompasses a combination of reduced responsiveness by insulin-sensitive cells and defects in insulin secretion [1,2].

Approximately 537 million people worldwide have diabetes mellitus [3]. The number of people living with type 2 DM has dramatically increased in all countries irrespective of socioeconomic status. These numbers do not include those living with type 2 DM (around 240 million people) but who remain undiagnosed [2,3]. Recently, it has become evident that diabetic patients are more susceptible to infection with Coronavirus disease 2019 (COVID-19) and thus have an increased mortality rate [4,5,6]. COVID-19 patients could also develop diabetes due to sedentary lifestyle and weight gain caused by lockdown and restricted movement [7,8]. Aside from the life-threatening risks associated with COVID-19, 6.7 million deaths are directly attributed to DM or its complications in 2021, with 541 million adults having impaired glucose tolerance [3,9]. The latest report from the World Health Organisation states that by 2030, this metabolic pandemic will become the 7th main cause of worldwide mortality as the prevalence is estimated to encompass more than 400 million sufferers [10,11]. This number has since risen in post-COVID estimates to 643 million by 2030 and 783 million by 2045 [3]. It is estimated that the cost for DM worldwide was USD $850 billion in 2017, and reached USD $966 billion in 2021 [3].

The exact aetiology of type 2 DM is yet to be confirmed; however, it appears to arise mainly from a complex polygenic and heterogeneous group of metabolic defects involving inadequate insulin secretion from β-cells following an inability of the insulin-sensitive tissues, such as adipose, skeletal muscle and cardiac muscle cells to adequately respond to increased blood glucose levels [12,13,14]. Several risk factors have been associated with the development of type 2 DM, which can be broken down into modifiable (hypertension, lack of physical activity and obesity) and non-modifiable (aging, being male or of particular ethnicity) risk factors [15].

The onset of type 2 DM occurs gradually over several years and patients often do not notice any symptoms [16]. In the early stages of pre-DM, the pancreatic β-cells overproduce insulin leading to hyperinsulinemia in order to compensate for the high glucose levels appearing in the circulating blood, thus giving normal or slightly above normal blood glucose readings. After a few years of this constant compensation, most pancreatic β-cells become damaged. Yet, during this period, some patients could restore some of their β-cell capacity with proper treatment and management [15].

After type 2 DM sets in, the disease continues to evolve with short or long-term complications that can occur at any stage of the disease progression. These include metabolic dysfunction or failure of several organs (kidney, heart, eyes and blood vessels), which increase the risk of developing cardiovascular diseases, such as hypertension, dyslipidaemia and obesity; this reduces the quality of life and increases the mortality rate among patients with type 2 DM [1,17,18,19,20]. Serious imbalance in blood glucose levels could result in serious organ dysfunction and death [21].

Glucose is the main source of energy for most body organs and tissues [22]. Following a meal, the body breaks food down into glucose and then transports it through the blood circulation. Under normal physiological conditions, increased blood glucose levels trigger the pancreatic β-cells of the islet of Langerhans to secrete insulin which facilitates glucose uptake, transportation and metabolism in peripheral tissues in order to maintain blood glucose within steady state levels (a process known as glucose homeostasis) [21,23]. Undeniably, insulin plays a major balancing role as a key hormone in glucose homeostasis and is the only one known to actively lower high blood glucose levels.

At present, treatment objectives in type 2 DM are focused on maintaining normal blood glucose levels. Different therapeutic approaches for type 2 DM are utilised. Changes in lifestyle, such as diet and exercise as part of a weight control program, are the first approach, and usually accompanied by pharmacological therapy [19]. Several treatment strategies are available for diabetic patients either through synthetic drugs or medicinal plant products from natural sources.

The currently available pharmacological treatments to help manage type 2 DM are well-established [24,25,26]. These medications can be divided based on their mechanism of action into three groups. The first group improves insulin sensitivity: the thiazolidinediones, also known as glitazones, including drugs such as pioglitazone and rosiglitazone [27]. The second group increases endogenous insulin: the glucagon-like peptide-1 (GLP-1) agonist [28], sulphonylureas, [29], dipeptidyl peptidase IV inhibitors [30] and alpha glucosidase inhibitors [31]. Finally, the third group mainly has one representative which is metformin, and this medication decreases hepatic production of glucose and reduces glucose intestinal absorption which directly affects insulin. Metformin is currently the first line therapy for type 2 DM [32].

Whilst most of these conventional antidiabetic drugs are potent and effective, they are not sufficient nor can they sustain optimal treatment for every diabetic throughout the ongoing progressive stages of the disease [33]. This is further complicated by the need to switch medications when side effects arise. For instance, lactic acidosis can occur in diabetic patients taking metformin, as well as being not recommended for patients with renal insufficiency [34,35,36]. Uncontrolled weight gain can occur with patients using sulphonylureas while thiazolidinediaones could increase the incidence of hyperlipidaemia [37]. Given the high cost of insulin, which may be required in combination to offset the limited capabilities of the oral anti-diabetic medications, the search for new drug entities from medicinal plants has become even more urgent.

Apart from prescribed medications, the vast majority of diabetic patients in developing countries use complementary or medicinal natural products that have an antidiabetic reputation [38,39]. Interestingly, most people at high risk of developing type 2 DM usually have the nutritional recommendation to take plant-based food, all known to be enriched with several phytochemicals, such as phenolic compounds [39]. Constituents of medicinal plants continue to provide some lead active compounds that can be further developed into antidiabetic medication [40]. Thus, medicinal plants continue to provide a potential cache of novel agents from natural products that could help in the normalisation of blood glucose levels.

Throughout modern history, chemical lead constituents from ancient natural products have helped in developing synthetic antidiabetic drugs [38]. For example, the discovery of metformin (1,1-dimethylbiguanide, Figure 1b) resulted from the isolation of guanidine and galegine (Figure 1a) from the medicinal plant *Galega officinalis* L. [41]. Initial testing indicated that guanidine was too toxic for clinical use, and thus attention turned to the less toxic constituent, galegine. Synthetic derivatives of these compounds led to the generation of the biguanide, metformin (Figure 1b) which was approved for use in the USA in 1995 [42].

At present, almost 25% of all pharmaceutical drugs on the market have originated from natural products [43]. In additional, although there are more than 250,000 medicinal plants with potential antidiabetic activity, only 1% have been pharmacologically investigated [44]. Therefore, there is great potential to discover new compounds from natural sources as an alternative prevention and treatment approach that is affordable and with possible low side effects to reduce the burden of this epidemic disease. Hence, research that targets medicinal plants for discovery of lead drug entities to treat or alleviate symptoms of type 2 DM is very promising: Table 1 outlines a few plants reported to exert effects via promoting insulin secretion. Plants that exert their glucose lowering effects via other activities are summarised by Patel et al. (2012) [45]).

Although the publications on the general anti-diabetic properties of *Teucrium polium* (*Tp*) were summarised by Asghari et al., a detailed analysis of the extract in relation to type 2 DM is not presented [46]. Phenolic compounds are of rising interest. However, these compounds are quickly metabolised after ingestion in the gut and via the first pass effect. Therefore, our current review aims to evaluate the literature on the use of *Tp* for treating type 2 DM and to discuss the assessment strategies of studies that examined key phenolic compounds identified in aqueous *Tp* extracts.

## 2. *Teucrium polium*

*Teucrium polium* L. (Lamiaceae) (*Tp*), commonly known as golden germander, is a shrubby plant mainly growing in the Mediterranean deserts, hills, and mountains. This plant forms a highly influential component of Middle Eastern and Palestinian folk medicine where the natives drink the plant extract as a tea to treat DM, rheumatism, gastrointestinal disturbances, and inflammations. The folk of that region are also reported to use *Tp* brew for its antihyperlipidemic, analgesic, diaphoretic and antipyretic activities [39,55]. Although there have been several studies that examined the total plant extract for its glucose lowering effect [46,47,48,49,55] and others that have identified several phytochemical constituents of the extracts that include flavonoids, terpenoids and iridoids [56,57], the exact compound(s) responsible for the glucose-lowering effect of the total extract of *Teucrium polium* are yet to be determined. A few cases showing the toxicological potential of *Tp*, especially hepatotoxicity, have been reported; however, it is not clear which plant species was consumed, nor whether additional medications contributed to the adverse outcomes [58,59].

Regrettably, safety evaluation of medicinal plants is difficult due to the complex chemical nature and diversity within plants, some of which are currently considered useful as complementary medicine in diabetes. These plants may contain thousands of unknown bioactive compounds or other contaminants and toxic materials. While broad toxicity profiles provide a measure of safety for such complex natural components, robust clinical and experimental data, including toxicological information, phytochemical properties and the safe use of reputed medicinal plants and their likely active constituents with potential antidiabetic effects, are urgently needed [60,61].

### 2.1. Antidiabetic Effect of Tp

Few ethnobotanical [56,62] or pharmacological [46,55,63,64] studies have focused on the glucose lowering properties of traditional medicinal plants, especially *Tp.* The antihyperglycemic activity of extracts of *Tp* have been studied in vivo, primarily in rodents, since the 1980s [50,55,63,64,65]. Gharaibeh et al. (1988) demonstrated that in both streptozotocin (STZ)-induced diabetic and normoglycemic rats, an aqueous extract of *Tp* aerial parts results in a substantial reduction in blood glucose levels 4 h after intravenous administration and 24 h after intraperitoneal administration [65]. Furthermore, chronic (30 consecutive days) oral administration of *Tp* extract at 0.5 mg/kg to STZ-induced diabetic rats significantly (*p* < 0.05) reduced blood glucose levels compared to the STZ-induced diabetic controls [63,64]. Unfortunately, these studies ran for an extended period and did not include a positive control, so results may need further analysis to confirm their data. Nevertheless, two animal studies have also reported an acute antihyperglycemic effect with use of this extract [50,55]. Oral administration of *Tp* extract at (125 mg/kg) to normoglycemic rats exhibited significant acute diminution of blood glucose levels 4 h after a single dose, and it was superior to glibenclamide [50].

In one of the more recent studies, Ireng et al. (2016) reported that intravenous administration of *Tp* extract (100 mg/kg) to normoglycemic rats, lead to a significant acute hypoglycaemic effect in a manner that was similar in efficacy to insulin over the first 30 min [55]. This outcome corroborated earlier results revealing that *Tp* extract could be able to ameliorate glucose homeostasis either by increasing insulin secretion or improving glucose uptake from peripheral tissues. Some studies have attempted to assess the plant extract further ex vivo and in vitro in an attempt to elucidate the possible mechanistic role of constituents with glucose lowering potential [48,64].

In an investigation by Mirghazanfari et al. (2010), *Tp* methanolic extract (1000 μg/mL in 2.8 mmol/L glucose) was able to potentiate (*p* < 0.05) glucose stimulated insulin secretion (GSIS) from isolated perfused rat pancreas ex vivo; however, no effect was observed with the aqueous extract [49]. Moreover, the insulinotropic effect observed was credited to the presence of flavonoids, including apigenin. In another study by Yazdanparast et al. (2005), ethanol/water extract of *Tp* at 1 to 100 μg/mL in 2 and 16 mmol/L glucose two hours after treatment was able to stimulate (*p* < 0.001) GSIS from isolated pancreatic rat islets; however, when tested at 1000 μg/mL, the effect was decreased [64]. This was attributed to the cytotoxicity of the plant extract. Moreover, studies that also histologically examined the islets of Langerhans of STZ-treated rats reported that the pancreatic islets are regenerated after *Tp* treatment which may be attributed to the quercetin in the extract [63,64,66].

The potential mechanism explaining the molecular pathways that could contribute to the glucose-lowering activity of the extract in cell culture settings have been studied as well. Relevant emerging in vitro studies have focused on insulin secretagogue potential from *Tp* extract [47,48]. Stefkov et al. (2011) evaluated the insulinotropic effect of *Tp* extract using INS-1E rat insulinoma cells and reported that *Tp* ethanolic extract at 500 μg/mL in 20 mmol/L glucose exhibited a significant increase in GSIS [50]. It is worth mentioning that in these experiments the cell viability assay was not measured and the causes of diminished activity at a higher dose were not clearly explained.

More recently, Mannan (2017), assessed the insulin secretagogue potential of the “aqueous” *Tp* extract, prepared as per methodology outlined in Ireng et al. (2016) [55], on BRIN-BD11 rat pancreatic cells. It was shown that the extract at 62, 125, 250 and 500 μg/mL in 5.5 mmol/L glucose significantly (*p* < 0.05) increased glucose uptake and insulin secretion in a dose-dependent manner [48]. Along with this, treatment with the *Tp* extracts significantly (*p* < 0.05) increased the expression of GLUT2 and glucokinase activity. This was associated with an increase in ATP production and an increase in influx of intracellular calcium [48]. In keeping with this finding, a study by Kasabri et al. (2012) reported distinct results in the increasing pattern of GSIS at different concentrations (10, 100, 1000, 10,000 or 25,000 μg/mL in 5.6 mmol/L glucose) of aqueous *Tp* extracts in mouse pancreatic β-cells MIN6 without existing cytotoxicity. Moreover, the secretory mechanism was highly dependent on calcium influx [47]. The differences in the pattern of GSIS could be due to different reasons including the contents of culture medium, growth rate, conditions or the type of cell line.

Since *Tp* extracts were able to reduce blood glucose levels by inducing insulin secretion in accordance with the well-known biochemical pathway for insulin secretion from pancreatic β-cells (Table 2), they are considered a promising initial target for drug discovery research, especially for DM. Although *Tp* antidiabetic activity has been studied in different experimental models, all of which support the use of its extracts as an initial phase in the search for antihyperglycaemic agents, the bioactive constituent/s of the *Tp* extract are yet to be defined.

### 2.2. Identified Bioactive Constituents in Tp

There are different secondary metabolites present in *Tp* extracts, such as diterpenes, essential oil and phenylethanoid glycosides, and among the differentially identified compounds the most prominent active constituents are the phenolic compounds, especially, the flavonoids [55,67,68,69,70]. Variations in the constituents reported between studies could be due to several factors including the method of chemical extraction used (maceration or ultrasonic extraction), type of solvent (polar or non-polar) and the detection accuracy of each separation technique applied, which varies significantly based on the type of instrument (UV, HPLC or LC-MS), and undoubtedly would affect the extract and compounds stability. The physico-chemical properties of the plant matrices are immensely complex, creating the possibility of these constituents existing in the plant extract in unstable form or even in very low concentrations. Consequently, this could contribute to an even more critical dilemma during the separation process when the activity of such bioactive compounds could diminish. Hence, choosing an appropriate technique is a very challenging task but significant when isolating and quantitating such bioactive entities.

Based on the literature, several phytochemical investigations have suggested that certain flavonoids, namely, apigenin [48,49,54,67,70,71,72,73], apigenin 7-glucoside [49,74], rutin [54,67,68,70,71] and quercetin [54,70], have glucose lowering potential and therefore may account for glucose lowering potential of a plant extract [48,49,65,71,72,73]. However, their quantity in any plant or extract, including *Tp* extracts, is yet to be determined and thus their contribution to the glucose lowering potential of any given plant material or extract activity is yet to be clarified. In plants, flavonoids and phenolic acids predominantly exist as conjugates of glucose and other glycosides that can be complex for enzymatic hydrolysis (e.g., linked via beta linkage or at inaccessible sites for human enzymes) [74]. Therefore, humans rely heavily on intestinal microflora to help in removing these glycosides and thus enable free flavonoids/phenolic acids (aglycones) to be absorbed through the intestines [75,76,77].

#### Bioactivity and Bioavailability of the Key Flavonoids Found in *Tp*

Apigenin (Figure 2) is a major flavone that can be extracted from different plant sources [78]. This flavone also exists in various plants in the form of sugar conjugates, such as apiin (obtained from celery and parsley), apigetrin (also known as apigenin-7-glycoside, derived from dandelion coffee) and vitexin (present in bamboo leaves) [79]. Identification of apigenin in *Tp* was reported in several publications in its aglycone form [49,50,55,72,80,81] and as a sugar conjugate, apigenin-7-*O*-glucoside or apigetrin [50,59,74].

Apigenin has been shown to have antidiabetic properties in different experimental settings (Table 3); however, these studies are limited. Chronic and acute administration of apigenin at both 4 and 25 mg/kg in diabetic rats caused a decrease in blood glucose levels [73] with a comparable effect to glibenclamide [73]; these studies and others are summarised in Table 3. Although some studies used antidiabetic drugs such as glibenclamide and glipizide as positive controls [73,82], the route of administration and doses were not comparable when proposing their significant results. The hypoglycaemic effect, as well as the proposed biochemical mechanism of apigenin in in vitro studies, were also assessed. In isolated pancreatic islets from non-diabetic and STZ-treated diabetic rats, Esmaeili and Sadeghi (2009) observed a significant increase in GSIS in apigenin-treated diabetic islets at 50 and 75 μg/mL in 5 or 11.1 mmol/L glucose, respectively. Nonetheless, no statistically significant effect was observed in the normal-treated group [81]. In a subsequent study by Stefkov et al. (2011), insulin secretion by the glycosidated form of apigenin, apigenin-7-glycoside at 500 μg/mL in 20 mmol/L glucose in INS-1E cells, was evaluated. It appears that this flavone caused an increment in GSIS [50]. According to these data, apigenin antidiabetic effects may be achieved as a result of insulinotropic properties. Importantly, however, the doses were used in high amounts compared to what actually might be in the total *Tp* extract.

Quercetin is a flavonoid naturally found in glycoside forms, such as rutin (quercetin 3-rutinose), as shown in Figure 3 [76]. Moreover, similar to the above discussed apigenin, both quercetin and rutin have been identified in *Tp* extracts [50,55,71,72,80,81]. Noticeably, the yield of these compounds in the extract from the original plant material is not known and therefore it is difficult to know whether these observations were due to an actual representation of these plant constituents or others.

Rutin and quercetin have been shown to have antidiabetic activity in various experimental models [66,81,83,84,85]. Studies (some outlined in Table 4) have shown that a reduction in blood glucose levels was significant in STZ-treated diabetic rats when treated orally with rutin [73,84]. In addition, rutin significantly (*p* < 0.05) increased plasma insulin levels in normoglycemic rats; therefore, no effect was observed when comparing the results from STZ-treated to healthy controls. In this study, the authors suggested a protective role of rutin on the β-cells by preventing STZ-induced oxidative stress, thereby increasing insulin secretion in the diabetic group [84]. Quercetin was able to reduce hyperglycaemia in Alloxan-induced diabetic mice via the oral route [83] and in STZ-induced diabetic rats injected with quercetin interparentally [64]. The mechanistic role of quercetin in insulin secretion was further investigated in pancreatic β-cells [50]. Stefkov et al. (2011), showed that treatment with either rutin or quercetin appeared to increase GSIS sensitivity in INS-1E rat insulinoma cells [50]. In another investigation, rutin was able to increase GSIS in isolated rat pancreatic islets from normal and STZ-exposed diabetic rats; however, no effect was observed in the normal control group [82]. In a subsequent study, quercetin was able to augment both glucose and glibenclamide stimulated insulin secretion in INS-1E; the mechanism was thought to be through the potentiation of the extracellular signal-regulated kinase 1/2 signaling pathway as a protective mechanism against oxidative stress [85]. Nonetheless, as the dosage used was very high, it may have produced statistically significant results but is unlikely to be transferable to physiologically relevant situations. Table 4 summarises a few studies related to quercetin and rutin including some cell culture experiments with quercetin which suggest that blood glucose levels are maintained via such mechanisms, including increased glucose uptake via muscle and liver cells for utilisation or storage [83,86].

Since ethnopharmacological data acknowledges that *Tp* is being traditionally brewed and consumed like tea, it is important to further acknowledge that it undergoes several biotransformations in the gut and the liver. Therefore, it is important to understand the possible different metabolites of the flavonoids that are generated upon ingestion of the extract. Bioavailability of flavonoids can be influenced by many factors including enzymatic hydrolysis and metabolic conjugation [87,88]. Metabolism of flavonoids starts from the mouth where some flavonoid glycosides are deglycosylated by saliva to give the flavonoid aglycone [87]. However, others undergo enzymatic hydrolysis and absorption in the small intestine [88,89] limiting absorption to about 5–10% [77].

Further to their hydrolysis, the majority of ingested phenolic compounds and flavonoids are found in the blood conjugated with glucuronic acid in which the conjugation site/pattern mainly depends on the molecular structure of the flavonoids [90]. Metabolites of apigenin, namely glucuronides and sulfonates, have been reported to be present in both rats and humans [77,91]. Metabolism of quercetin presumes formation of mixed conjugates. Primary quercetin metabolites appear to be quercetin 3-*O*-glucuronide, quercetin 3′-*O*-sulfate, isorhamnetin and quercetin glutathione conjugate [92,93].

It is noteworthy that each different class of active constituents isolated from *Tp* extracts (e.g., flavonoids) could exist in multiple forms. Further, these compounds are metabolized in the gut and by the liver to give a range of biological metabolites. Therefore, studies thus far that have assessed flavonoids in their unconjugated form or directly from extracts must be evaluated with caution and future efforts should consider the final metabolites circulating in the body [91,92,93,94].

## 3. Conclusions

Type 2 DM is a multifactorial disease that requires a comprehensive treatment approach. Considering the prevalence of type 2 DM and current treatment strategies, medicinal plants with antidiabetic activity, such as *Tp*, are a valuable source of research for alternative hypoglycaemic agents. Pharmacological studies have repeatedly confirmed the hypoglycaemic effect of *Tp* extract in vitro and in vivo. The active constituents of the extract are not well defined from the scientific literature; however, it might be related to the presence of flavonoids.

Few independent studies demonstrate the glucose-lowering potential of apigenin, quercetin and rutin. Some results from in vitro studies suggest that *Tp* extracts increase GSIS by activating different well-known insulin signalling pathways in pancreatic β-cells and the presence of these flavonoids may have potential to convey, at least in part, a similar effect at quantities in the plant extract. However, the observed glucose-lowering potential of these flavonoids has currently been based on the activity of individual compounds used at relatively high doses and, most importantly, their concentrations in the plant and their ultimate concentrations in the blood, post ingestion, are not known. The uncertainty in these amounts within the overall plant extract could present a caveat to the validity of the glucose related observations reported. Further, these compounds are quickly metabolised in the gut and post-absorption, generating glucuronide/sulfonate/methylated forms, i.e., they do not appear as aglycones in the blood stream, and thus may present with different activities and potencies to currently studied aglycones (or glycosides from direct exposure to the extract.

Toxicological studies examining *Tp* extract are limited. Although *Tp* extract was reported to induce liver toxicity, the reported studies are not conclusive and thus further assessments are needed. In any case, the antidiabetic effects of the extract are most likely not related to the same compounds driving the reported liver toxicity; however, even if they were, drug development techniques and knowledge about pathological mechanisms of hepatotoxicity would ameliorate and prevent such side effects.

Insulinotropic activity is another aspect of glucose metabolism that is currently lacking from the literature, with a paucity of information regarding the effectiveness of the flavonoids within *Tp*. Thus, further investigations aiming to elucidate the mechanistic role of some of the chemical constituents conveying the insulin secretagogue characteristics of the *Tp* extract are essential, as we move towards a deeper understanding of the molecular mechanism cascades, ultimately assisting in the development of new effective hypoglycaemic agents, especially for type 2 DM.

## Figures and Tables

**Figure 1 biology-11-00128-f001:**
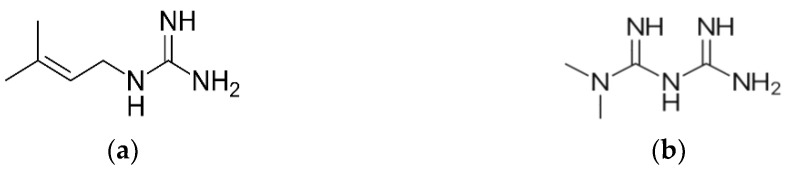
Structures of (**a**) Galegine and (**b**) Metformin.

**Figure 2 biology-11-00128-f002:**
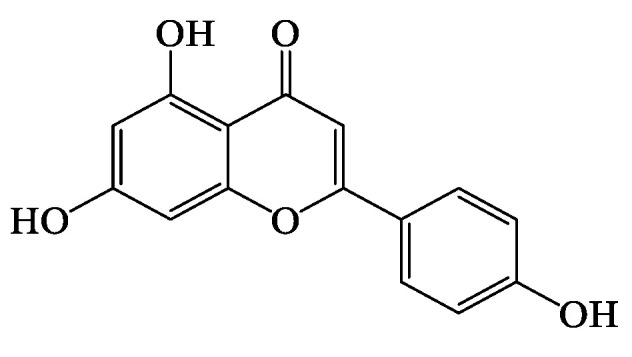
Structure of Apigenin.

**Figure 3 biology-11-00128-f003:**
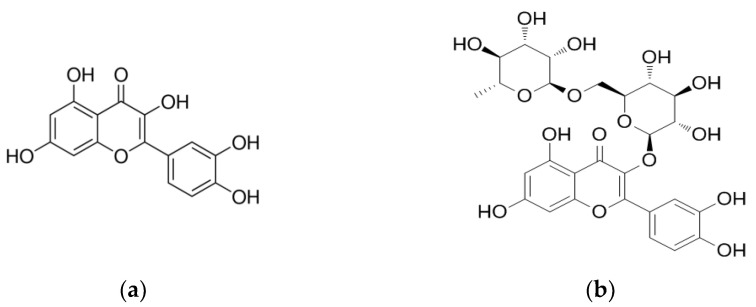
Structures of (**a**) Quercetin and (**b**) Rutin.

**Table 1 biology-11-00128-t001:** Some antidiabetic medicinal plants with potential insulin secretagogue activity.

References	Plant Name	Antidiabetic Activity
[46,47,48,49,50]	*Teucrium polium*	Increase insulin secretion
[51]	*Tabernaemontana divaricata*	Increase blood insulin and promote pancreatic β-cells regeneration in mice
[52]	*Gymnema sylvestre*	Increase insulin and pancreatic β-cells regeneration in rodent
[53]	*Ficus deltoidea*	Increase insulin secretion
[54]	*Bidens pilosa*	Increase insulin secretion

**Table 2 biology-11-00128-t002:** Studies on the insulinotropic effect of Tp on pancreatic cells/tissues.

Reference	Design	Target	Extract	Treatment	Possible Mechanistic Role
[47]	In vitro	Pancreatic β-cells MIN6	Aqueous	*Tp* at 10, 100, 1000, 10K & 25K μg/mL in 5.6 mmol/L glucose) and L-alanin (10 mmol/L) as a positive control	↑ GSIS (25K μg/mL was most effective), mainly via Ca ^2+^ influx as this effect reduced when tested in calcium-free-KRH buffer (−2.5 mmol/L CaCl_2_)
[48]	In vitro	Pancreatic β-cells BRIN-BD11	Methanol	*Tp* at 62, 125, 250 & 500 μg/mL in 5.5 mmol/L glucose) and Gliclazide (5, 500 μmol/L in 5.5 mmol/L glucose) as a positive control	↑ GSIS, and glucose uptake (*p* < 0.05) through: ↑ GLUT2 expression, ↑ Glucokinase, ↑ ATP production, ↑ Ca ^2+^ influx
[49]	In situ	Isolated perfused rat pancreas	Methanol	*Tp* at 1000 μg/mL in 2.8 or 16.1 mmol/L glucose	↑ GSIS, mainly via Ca ^2+^ and K ^+^ channel as this effect reduced when tested in the presence of diazoxide and verapamil. Notably, apigenin was the only bioactive constituent detected by GCMS analysis
[50]	In vitro	Pancreatic β-cells INS-1E	Ethanol	*Tp* at 5, 50, 250, 500 & 1000 μg/mL in 20 mmol/L glucose	↑ GSIS in a dose-dependent manner (500 μg/mL was most effective; however, at 1000 μg/mL this effect was decreased

*Tp*: *Teucrium polium* extract; GSIS: Glucose-stimulated insulin secretion; GCMS: Gas Chromatography Mass Spectrometer; Ca ^2+^: Calcium; K ^+^: Potassium; GLUT2: Glucose transporter 2; ATP: Adenosine triphosphate, ↑ increased.

**Table 3 biology-11-00128-t003:** Apigenin potential in lowering blood glucose levels.

Reference	Target	Study Design	Dose	Positive Control	Duration	Outcome
[50]	Pancreatic β-cells INS-1E	Group 1: Glucose only at (20 mmol/L) Group 2: Apigenin-7-glycoside	(500 μg/mL in 20 mmol/L glucose)	-	30 min	Significant (*p* < 0.05) increase in GSIS compared to 20 mmol/L glucose
[72]	STZ-induced (40 mg/kg i.p) diabetic rats (150–250 g)	Group 1: Normal control received saline + 5% Ethanol *n* = 6 Group 1: Diabetic control received saline + 5% Ethanol *n* = 6 Group 2: Diabetic + Apigenin *n* = 6 Group 3: Diabetic + Apigenin-7-glycoside *n* = 6 Group 4: Normal group received no treatment *n* = 6	Apigenin i.p at (4 mg/kg/day)	-	7 days	On day 7, apigenin significantly (*p* < 0.01) reduced blood glucose levels compared with diabetic control. No significant effect with apigenin-7-glycoside.
[73]	Alloxan-induced (65 mg/kg i.v) albino diabetic rats (150–250 g)	Group 1: (3% Tween 80, 5 mL/kg) as diabetic control *n* = 6 Groups 2–4: Apigenin *n* = 6 each Group 5: Glibenclamide at (5 mg/kg) *n* = 6	Apigenin orally at (25, 50 and 100 mg/kg)	Glibenclamide at (5 mg/kg)	1 day	Significant (*p* < 0.05) reduction in mean glucose level compared to diabetic control. At 25 mg/kg, hypoglycaemic effect of apigenin was comparable to that of Glibenclamide (*p* < 0.05)
[73]	Normoglycemic rats (150–250 g)	Fasted normoglycemic rats treated as: Group 1: (3% Tween 80, 5 mL/kg) as normal control *n* = 6 Group 2: Apigenin at (25 mg/kg) *n* = 6 Group 3: Apigenin at (50 mg/kg) *n* = 6 Group 4: Glibenclamide (5 mg/kg) *n* = 6	Apigenin orally (25 and 50 mg/kg) Adrenaline (0.8 mg/kg i.p) injected after 2 h of treatment	Glibenclamide at (5 mg/kg)	4 h	Significant dose-related decrease in hyperglycemia response to adrenaline compared to that of normal control (*p* < 0.05). The effect of apigenin at 25 mg/kg was comparable to that of Glibenclamide. A blood sample was taken at: 30, 60, 90, 120, 180 and 240 min
[73]	Normal and alloxan-induced (65 mg/kg i.v) albino diabetic rats (150–250 g)	Glycogen content (from skeletal muscle and liver tissues) Group 1: Normal non-diabetics rats (3% Tween 80, 5 mL/kg/day) as normal control *n* = not given Groups 2: Alloxan-induced diabetic rats + 3% Tween 80, 5 mL/kg/day) as diabetic controls *n* = 6 Group 3: Alloxan-induced diabetic rats + Apigenin at (50 mg/kg/day) *n* = 6 Group 4: Alloxan-induced diabetic rats + Glibenclamide (5 mg/kg/day) *n* = 6	Apigenin orally at 50 mg/kg/day	Glibenclamide at (5 mg/kg)	7 days	Apigenin gave a significant (*p* < 0.05) reduction in fasted glucose level compared to diabetic control. Liver and muscle glycogen content significantly (*p* < 0.05) increased with apigenin.
[81]	Isolated islets from normal and STZ-induced diabetic rats	Group 1: Normal control Group 2: Normal + Apigenin Group 3: Diabetic control Group 4: Diabetic + Apigenin	Apigenin at 50 or 75 μg/mL in 5 or 11.1 mmol/L glucose, respectively	-	30 min to 3.5 h	Significant (*p* < 0.05) increase in GSIS in islets from apigenin treated STZ-diabetic rats compared to the STZ-diabetic controls.
[82]	STZ-induced (40 mg/kg i.p) diabetic male rats and normoglycemic rats (110–130 g)	Group 1: Normal control (0.1% *v*/*v* DMSO via i.p every other day) Group 2: STZ-diabetic control Group 3: STZ- diabetic + Apigenin Group 4: Apigenin control Group 5: Glipizide control Group 6: STZ-diabetic + Glipizide	Apigenin i.p (1.5 mg/kg) every alternate day and Glipizide orally (5 mg/kg) daily, for 28 days; starting from day 15 after the STZ injected	Glipizide orally (5 mg/kg) daily	43 days	FBG levels measured every 7 days. Apigenin significantly (*p* < 0.05) decreased blood glucose levels. enhanced GLUT4 translocation, decreased CD38 expression and preserved β-cell distruction.

STZ = Streptozotocin; i.v: Intravenous; i.p: Interparental; *n*: Number; *p*: *p* value; GSIS: Glucose-stimulated insulin secretion; DMSO: Dimethyl sulfoxide, GLUT4: Glucose transporter 4, FBG: Fasting blood glucose.

**Table 4 biology-11-00128-t004:** Quercetin and rutin potential in lowering blood glucose levels.

Reference	Target	Study Design	Dose	Duration	Outcome
[50]	Pancreatic β-cells INS-1E	Group 1: Glucose only (20 mmol/L) Group 2: Rutin Group 3: Quercetin	Rutin or Quercetin (500 μg/mL in 20 mmol/L glucose).	30 min	Significant increase in GSIS compared to 20 mmol/L glucose (*p* < 0.05).
[72]	STZ-induced diabetic rat (150–250 g) at (40 mg/kg i.p)	Group 1: Diabetic control (Saline + 5% Ethanol)—*n* = 6 Group 2: Diabetic + Rutin—*n* = 6 Group 3: Normal control—*n* = 6	Rutin i.p at (4 mg/kg)	7 days	Rutin significantly (*p* < 0.01) reduced blood glucose levels (oral glucose tolerance test) compared to the diabetic control.
[81]	Isolated islets from normal and STZ-induced diabetic rats	Group 1: Normal control Group 2: Normal + Rutin Group 3: Diabetic control Group 4: Diabetic + Rutin	(50 and 75 μg/mL in 5 or 11.1 mmol/L glucose)	30 min to 3.5 h	Significant (*p* < 0.05) increase in insulin secretion in rutin treated islets from STZ-induced diabetic rats compared to STZ-induced diabetic control.
[83]	Swiss albino mice Alloxan-induced diabetic (150 mg/kg i.p) and normal rats	Group 1: Normal (saline) control (*n* = 6) Group 2: Alloxan-diabetic rats (*n* = 6) Group 3: Alloxan-diabetic + quercetin (*n* = 6) For GLUT4 (Serum and tissue homogenates adipocytes and skeletal muscles)	Orally (quercetin (20 mg/kg/day)	3 weeks	↓ FBG (*p* < 0.05) ↑ Hexokinase (*p* < 0.05) ↓ FBPase (*p* < 0.05) ↓ G6Pase (*p* < 0.05) ↑ GLUT4 (*p* < 0.05)
[84]	STZ-induced male rats (150–180 g) at (50 mg/kg i.p) and healthy rats	Group 1: Normal control (*n* = 8) Group 2–4: Normal + Rutin (*n* = 8 each) Group 5: Diabetic control (*n* = 8) Group 6–8: Diabetic + Rutin (*n* = 8 each)	Orally (rutin at 25, 50 and 100 mg/kg), (1 mL/rat)	45 days	Significantly decreased the plasma glucose levels by the different doses (44.36%, 50.92% and 62.73% respectively) compared to diabetic control (*p* < 0.05). At 100 mg/kg, rutin significantly increased plasma insulin level by 58.49% (*p* < 0.05).

STZ: Streptozotocin; i.p: Interparental; GLUT4: Glucose transporter 4; FBG: Fasting blood glucose; i.p: Interparental; AMPK: Adenosine monophosphate-activated protein kinase; G6Pase: Glucose-6-phosphatase; FBPase: Fructose-1,6-Bisphosphatase; *n*: Number. ↑ Increased, ↓ Decreased.

## Data Availability

This is a review and thus relied on peer reviewed published research.

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
