# Peer review of "Teucrium polium: Potential Drug Source for Type 2 Diabetes Mellitus"

_biology, 2022, doi:10.3390/biology11010128_

Round 1
Reviewer 1 Report
The paper entitled “The insulin secretagogue effects of Teucrium polium; potential drug source for type 2 diabetes mellitus” is informative and noteworthy.
I don`t have any additional comments.

Author Response
Thank you, we have spell checked and edited throughout the manuscript. Also, included are edits in accordance with other reviewers' comments.
Reviewer 2 Report
Dear Autors,
The article Albadr et al. describes The insulin secretagogue effects of Teucrium polium; potential drug source for type 2 diabetes mellitus. The problem of the prevalence of type 2 diabetes mellitus is very important. The search for new, interesting natural drugs is very important, the more so as the side effects of using synthetic drugs are becoming more and more frequent. The manuscript presented by the authors is interesting and introduces many new elements.
Minor error:
- I propose not to repeat the keywords from the title of the manuscript.
- In 'Introduction' underline the novelty element.
- At the beginning of section 2, write more about the chemical composition of Tp.
- Instead of 'Summary' section, there should 'Conclusion'. The name 'Summary' is confused with 'Abstract'.
- Adapt 'References' to the requirements of Biology MDPI.
- In 'References' use italics of Latin names.
General, the manuscript well written. I recommend publishing in Bilogy MDPI.
Author Response
Thank you for reviewing our manuscript, we provide the following responses to your comments:
- I propose not to repeat the keywords from the title of the manuscript.
- The title has been slightly modified and thus the key words modified to avoid duplication as well as align better with the aims and the paper.
- In 'Introduction' underline the novelty element.
- The novelty of the review is highlighted in the track edited manuscript attached, in the last paragraph of the introduction "lines 146 to 152".
- At the beginning of section 2, write more about the chemical composition of Tp.
- Instead of 'Summary' section, there should 'Conclusion'. The name 'Summary' is confused with 'Abstract'.
- The section has been renamed and slightly reworded.
- Adapt 'References' to the requirements of Biology MDPI.
- This has been adapted.
- In 'References' use italics of Latin names.
- This has been addressed

Reviewer 3 Report
The review brings interesting and relevant information about the anti-diabetic effects of extracts of Teucrium polium (Tp) against diabetes type 2. Despite being well written, in my opinion, the review needs some reorganization, in order to better present the main issue of the paper. For example, in the title, the authors focus on "insulin secretagogue effects of Tp extracts". However, along the review, they also present data about other anti-diabetic modes of action (different from action in insulin secretion). Besides, they also present information of the main compounds, possible responsible for the anti-diabetic effects of Tp extracts. Therefore, the authors should check carefully the information presented, in order to be in consonance with the title and aims; or adequate them both to the information presented. In addition, the authors must cite the following paper DOI:10.4103/2221-1691.290868 (Asian Pacific Journal of Tropical Biomedicine 2020; 10(10): 433-441), since it is a recent review also focusd on the antidiabetic effects of Tp, and make clear how their review is different from this previous one, since some of the information are the same. This is an important aspect. Besides, especially the part of the bioactive flavonoids should be rearranged (presenting firstly the flavonoids and then the bioavailabitiy aspects), also with some tables being merged. In addition, please check the suggested corrections in the PDF version (here attached).

Author Response
Thank you for your insightful comments, which we have attempted to address.
- The review needs some reorganization, in order to better present the main issue of the paper. ...
- We have attempted to address this by modifying the manuscript title, sharpening the aims and rearranging the sections as per your comments.
- We have cited the paper you proposed and articulated how our review is different (p4, lines 146-152).
- We have rearranged the relevant sections to enable a smoother flow and condensed the tables.
- Annotated comments on the manuscript have been addressed.
- Spell check conducted and fixed.
